# Correlation between Anthropometric and Ultrasound Measurement for Dry Needling of the Iliocostalis Lumborum Muscle with a Safety Protocol: A Cross-Sectional Observational Study

**DOI:** 10.3390/healthcare10122470

**Published:** 2022-12-07

**Authors:** Raquel López-Castellanos, Enrique Ruiz-Astasio, Antonio Cortés-Campos, Samuel Fernández-Carnero, Nicolás Cuenca-Zaldivar, Daniel Pecos-Martin, Francisco Selva-Sarzo, Susana Nunez-Nagy

**Affiliations:** 1Private Clinical Practice, 28522 Madrid, Spain; 2Universidad de Alcalá, Facultad de Medicina y Ciencias de la Salud, Departamento de Enfermería y Fisioterapia, Grupo de Investigación en Fisioterapia y Dolor, 28801 Alcalá de Henares, Spain; 3Research Group in Nursing and Health Care, Puerta de Hierro Health Research Institute—Segovia de Arana (IDIPHISA), 28222 Madrid, Spain; 4Department of Physiotherapy, University of Valencia, 46010 Valencia, Spain

**Keywords:** ultrasonography, iliocostalis lumborum, dry needling

## Abstract

Introduction: the management of musculoskeletal pain through the application of dry needling (DN) is effective. The application of this technique can carry very infrequent major risks on muscles, such as on the iliocostalis lumborum due to its proximity to the kidney and the peritoneum. It is important to establish a DN protocol based on the different anthropometric variables of the subjects. Main objective: the main objective of this study was to investigate the correlation between different anthropometric variables and the skin-kidney and skin-peritoneum distances to establish the size of the needle that could perform DN in the iliocostalis lumborum muscle without risk. Design: a cross-sectional observational study was conducted. Methodology: a total of 68 healthy subjects were evaluated. Demographic and anthropometric data, such as age, gender, weight, height, body mass index (BMI), chest (xiphoid process and axilla) and abdomen circumferences, and skinfold thickness were collected. The measurements of skin-upper and lower edge of the iliocostalis lumborum muscle and the skin-peritoneum and/or kidney in the regions of L2 and L4, and on both sides, were assessed using ultrasound imaging. Results: a multiple linear regression analysis was performed, confirming that, in L2 without compression, gender significantly predicted the distance, with the distance being greater in women than in men. The measurement without compression increased with age up to 50 years, and it also increased with higher measurements for the chest-triceps, iliac crest, and thigh skinfold thickness, and decreased with higher measurement for the abdominal circumference. It was verified that the measurement with compression in L2 decreased as the neutral axillary circumference and the skinfold thickness in the abdomen-iliac crest increased, while the distance increased with larger measurements obtained in the neutral abdominal circumference and in the skinfold thickness of the chest-triceps. It was also verified that the measurement with compression in L4 increased up to a body mass index of 25 and then decreased even if the index increased further, and it decreased as the skinfold thickness in the abdomen-iliac crest decreased and increased as the measurements of the neutral abdominal circumference and the skinfold thickness in the chest-triceps increased. In L4 without compression, the gender variable significantly predicted changes in the measurement, with women tending to have a smaller distance compared to men. Conclusions: the measurements of the neutral abdominal circumference, chest-triceps, and abdomen-iliac crest skinfold thickness could help clinicians predict the skin-kidney and skin-peritoneum distances for dry needling of the iliocostalis lumborum with the methodology described.

## 1. Introduction

Myofascial Pain Syndrome (MDS) is characterized by the presence of myofascial trigger points (MTrP) [1,2]. There are several approaches to MTrP, including Dry Needling (DN) [3]. Recent studies have demonstrated its effectiveness in the management of musculoskeletal pain and MDS [4,5,6]. The performance of DN entails associated risks and can provoke adverse effects [7]. Minor adverse effects can be found in the short term and cases of major adverse effects have been reported although they are unusual [8], such as pneumothorax, infection, excessive symptomatology exacerbation, septic arthritis, peripheral nerve injury, hemiplegia, coma, and even death [2,7]. Hall J.S. et al. claim caution when dealing with the abdominal or pelvic region. Some structures, such as the kidneys, peritoneum, or intestines, are contiguous to the peritoneal cavity and, therefore, a potential risk should be highlighted when this technique is performed in this region [9]. In this case, the authors add that special care should be taken when performing DN in the iliac, in the quadratus lumborum, and/or in the lumbar paraspinal muscles [9]. In addition, the kidney’s morphological variations should be taken into account [9].

To make DN a safe practice, Halle, J.S. and Halle, R.J. [8] conclude that it is advisable and necessary to educate and train professionals in anatomy, basic sciences, and clinical foundations. However, Boyce, D. et al. [7] found no association with regard to age, level of education, or years of DN experience. On the other hand, the lack of protocols or instructions to carry out DN safely in each muscle is evident. Currently, some authors report that a low body mass index (BMI), a narrow chest wall, and atrophy of the neck and chest muscles are risk factors to consider [10,11]. It has also been determined that, in obese subjects, it is dangerous because of the difficulty in estimating the depth to which the needle must penetrate [10]. Other important factors are the position of the patient [12] (depth from the skin to pleura varies) and the direction and angle of the needle, as well as its length [10].

Valera Calero, JA. et al. [13] set out safe instructions to perform dry needling on the rhomboid muscle and to avoid the risk of producing a pneumothorax.

Some authors already support the use of ultrasound imaging to access deep muscles or neurovascular structures more safely with the needle [2], not only to use eco-guided needling directly but also as a tool to assess and establish potential risks in terms of anthropometric measures depending on the characteristics of the subject [13,14,15].

Dry needling is a useful and daily application in the treatment of lumbar and thoracic pain, and iliocostalis lumborum is one of the goals in clinical setting with different patients [1,2]. The approach for this muscle is described in the manual by Travel and Simons [3]. Currently, several authors [2,13,15,16] recommend the application of DN accompanied by ultrasound imaging to avoid risks during its performance, but this tool is not available to all physiotherapists. As a result, there is a need to establish safer protocols to prevent serious adverse effects, such as those described in the previous section. In response to this need, this study investigated if different anthropometric variables can predict the depth of the kidney/peritoneum in order to propose a DN protocol in the iliocostalis lumborum muscle, since some cases have been described in the literature in which the performance of the DN technique in this muscle presents some risks due to its proximity to the peritoneum and/or kidney. In addition, no other study has been found in the literature that proposes a dry needling protocol on this muscle while taking into account the anthropometric measurements of the subjects with the help of ultrasonography.

The main objective of the study was to investigate the correlation between different anthropometric variables and the skin-kidney and skin-peritoneum distances to establish the size of the needle that could perform DN in the iliocostalis lumborum muscle without risk, by inserting the needle completely perpendicularly without the use of ultrasonography. The correlation between the anthropometric measurements (age, sex, height, weight, body mass index (BMI), chest (axillary and xiphoid process) and abdomen circumference, and skinfold thickness measurements) and the ultrasound scanner variables (distances of skin-kidney/peritoneum and of skin-muscle) was examined in this study to establish a predictive model.

## 2. Materials and Methods

### 2.1. Study Design

This was a cross-sectional observational study that attempted to establish a safe DN protocol in the iliocostalis lumborum muscle. This study followed the STROBE Statement [17,18].

### 2.2. Participants

#### 2.2.1. Population

This study included 68 subjects aged 28.26 ± 11.03 years old, with mostly women (54.4%) and a body mass index of 23.33 ± 3.05 (Table 1).

#### 2.2.2. Sampling

The recruitment of the sample was carried out through a non-probabilistic convenience sampling by posting posters at the University of Alcalá (UAH), where all the measurements were performed. The sampling and selection of participants are shown in a flowchart diagram in Figure 1.

#### 2.2.3. Sample

Based on previous similar studies, such as the Valera-Calero et al. studies [13,14] that described a sample size of ten subjects for each variable, the sample size was calculated to be 70 subjects. This method considered a range from 5 to 10 subjects per potential predictor, and, considering that the regression had 8 variables, the result would be 40 to 80 subjects.

#### 2.2.4. Eligibility Criteria

The subjects included in this study were healthy subjects between 18 and 65 years of age. The exclusion criteria included presenting chest pain in the last year; having a previous history of surgery in the lumbar-abdominal region; receiving pharmacological treatment that influenced muscle tone; presenting medical conditions that could modify the distance between the skin-kidney or skin-peritoneum, such as tumor; infection; and being a competitive athlete or having a high frequency of training (more than 5 days per week).

### 2.3. Ethical and Legal Aspects

This study was performed following all the guidelines approved by the Ethics Committee for Animal Research and Experimentation (CEI-EA) of the UAH, which is based on the Helsinki Declaration, and all personal data were processed in accordance with the Organic Law 3/2018, of December 5, of Protection of Personal Data and Guarantee of Digital Rights and the General Data Protection Regulation (EU) 2016/679.

### 2.4. Data Collection and Analysis

An assessment session was held at the Faculty of Nursing and Physiotherapy of the University of Alcalá which lasted approximately 30 min. All sociodemographic and anthropometric variables were collected to subsequently perform the ultrasound measurements.

#### 2.4.1. Collection of Sociodemographic and Anthropometric Measurements

The variables sex, age, weight, height, and BMI were collected by asking the subjects directly.

Chest and abdominal circumferences were performed with the subjects in a supine position with the arms in 90° shoulder abduction, and three measurements were taken: one at maximum inspiration, another at maximum expiration, and the last one in a neutral position with no associated abdominal movement at the axilla, xiphoid process, and abdomen levels (Figure 2).

For skinfold test, the method of Jackson, Pollock, and Ward (1978–1980) [19] was used to estimate body density following a normogram that was established by Baum and Raven, 1981.

#### 2.4.2. Collection of Ultrasonography Measurements

A Vinno model E35 ultrasound machine with a convex probe and a footprint of 40 mm was used as an exploration and measurement tool. Regarding the ultrasonography examination of the iliocostalis lumborum muscle, a prone position for the subjects was chosen [20] with the arms along the body, a pillow under the belly, and another pillow under the feet to place the pelvis in a neutral position. The researcher was positioned on the left side of the subjects. The side from which the data would start to be collected was randomized for all subjects. In addition, for the first 10 subjects, three repeated measurements of all the measurements were performed in order to obtain intra-observer reliability.

First, the last lumbar vertebra and the first sacral vertebra were located, and from there, the spinous processes of L4 and L2 were marked and assessed using ultrasonography. At the level of L2, the spinous process is located in the short axis, and the probe was moved to the right or left lateral side depending on randomization. We visualized the skin and the fatty tissue, we left the longissimus muscle towards the medial side, and we placed the iliocostalis lumborum muscle in the long axis, as well as the quadratus lumborum and the kidney, in the center of the screen. We asked the subjects to breathe in to see how the kidney descended and became more visible. When in maximum inspiration, we froze the image to make the following measurements: skin-upper edge of the iliocostalis lumborum muscle, skin-lower edge of the iliocostalis lumborum muscle, and skin-kidney. Then, a different researcher placed two fingers exerting pressure, as if it were the “tissue depression prior to performing the DP”. More ultrasound gel was placed between the fingers, and the sonographer placed the probe again in the short axis. An inspiration was requested to freeze the image and perform the same measurements mentioned above. At L4 level, the same protocol was followed with the difference in that, instead of performing the last skin-kidney measurement, the skin-peritoneum measurement was performed. The measurements were made on the right and left sides and with and without compression (Figure 3, Figure 4 and Figure 5).

### 2.5. Statistical Analysis

Statistical analysis was performed using the R Ver. 3.5.1 program. (R Foundation for Statistical Computing, Institute for Statistics and Mathematics, Welthandelsplatz 1, 1020 Vienna, Austria).

The level of significance was established at *p* < 0.005. The distribution of the quantitative variables was tested using the Kolmogorov–Smirnov test with Lilliefors correction, which showed the absence of normality.

Quantitative variables were calculated with mean ± standard deviation and categorical variables with absolute and relative values (%). The intra-observer ICC (2, 1) intraclass correlation coefficient was calculated as relative reliability, defined as poor (<0.5), moderate (0.5–0.75), good (0.75–0.9), and excellent (>0.9), and as absolute reliability, the standard error of measurement (SEM) was calculated. Likewise, the Mann–Whitney U test was applied between normal and reduced measurements. It can be verified that practically all the variables have a non-normal distribution except for the variables weight, height, body mass index, neutral axillary circumference, inspiration axillary circumference, expiration axillary circumference, expiration xiphoid circumference, inspiration abdominal circumference, expiration abdominal circumference, chest-triceps skinfold thickness, thigh skinfold thickness, and fat percentage (shown in red) (Appendix A).

Four regression models were constructed using the measurements with and without compression at L2 (skin-kidney distance) and L4 (skin-peritoneum distance) on the dominant side (since no significant differences were found with the Mann–Whitney U test between the dominant and non-dominant side measurements in the ultrasound). The predictor variables included gender, age, body mass index, neutral axillary circumference, neutral xiphoid circumference, neutral abdominal circumference, chest-triceps skinfold thickness, abdomen-iliac crest skinfold thickness, and thigh skinfold thickness. Since the assumption of linearity between all the dependent and predictor variables was not fulfilled, a generalized additive model (GAM) was applied with the double penalty method in the selection of variables. The concurrency of the variables was tested, removing those with a value greater than 0.8 from the models. The distribution of the residuals and of the fitted values around the null value and the adequacy of the number of basic functions with a non-significant K index were also checked.

## 3. Results

### 3.1. Descriptive Analyses

Table 1 shows the baseline data of the participants. The sample consisted of 68 subjects aged 28.26 ± 11.03 years, with mostly women (54.4%) and a body mass index of 23.33 ± 3.05.

### 3.2. Intra-Observer Reliability

It was verified that the ICC values (2, 1) are significant and have good to excellent values (Table 2).

### 3.3. Reliability with Compression vs. without Compression

The absence of significant differences between the measurements with and without compression was confirmed (Appendix A). It was checked that the ICC (2, 1) values are significant and have moderate to excellent values (Table 3).

### 3.4. Regression Analysis

#### 3.4.1. Dominant vs. Non-Dominant Side Comparison

The absence of significant differences between the dominant vs, non-dominant leg was verified, so the regression models were conducted with the measurements in the dominant leg (Appendix A).

Analysis of the dependent variable L4 without compression.

In the parametric model, the gender variable significantly predicts changes in the measurement without compression in L4; specifically, women tend to have a smaller distance compared to men (significant results are shown in red) (Table 4). In the smoothed model, the variables of body mass index, neutral axillary circumference, neutral abdominal circumference, triceps-chest skinfold thickness and iliac crest-abdomen skinfold thickness significantly predict the measurement without compression in L4 (significant results are shown in red) (Annex SIV). The measurement without compression in L4 increases with body mass index up to an approximate value of 25 and then decreases with higher indices, and it progressively decreases with higher values both in the neutral axillary circumference and in the abdomen-iliac crest skinfold thickness. It also progressively increases with higher values in the neutral abdominal circumference (especially from perimeters of 80 cm) and in the chest-triceps skinfold thickness.

#### 3.4.2. Analysis with the Dependent Variable L4 with Compression

The parametric model shows how gender does not significantly predict L4 compression measurements.

The variables of the smoothed model including body mass index, neutral abdominal circumference, triceps-chest skinfold thickness, and iliac crest-abdomen skinfold thickness significantly predict the measurement with compression in L4 (significant results are shown in red) (Table 5) The measurement with compression in L4 increases up to a body mass index of 25 and decreases even if the index increases further, and it decreases as the abdomen-iliac crest skinfold thickness decreases and increases as the neutral abdominal circumference (especially from a circumference of 80 cm) and chest-triceps skinfold thickness increase (Appendix A).

#### 3.4.3. Analysis with the Dependent Variable L2 without Compression

The parametric model shows how gender significantly predicts changes in the distance measured in L2 without compression; specifically, being a woman implies a greater distance compared to men (significant results are shown in red) (Table 6). The variables of the smoothed model including age, neutral abdominal circumference, triceps-chest skinfold thickness, iliac crest-abdomen skinfold thickness, and thigh skinfold thickness significantly predict the measurement without compression in L2 (significant results are shown in red) (Appendix A). The measurement without compression in L2 increases with age up to approximately 50 years, when it stabilizes and tends to decrease slightly over the years. It increases when the measurements of the chest-triceps, abdomen-iliac crest, and thigh skinfold thickness are greater, and decreases when the abdominal circumference is in a neutral range until approximately 80 cm, at which point it begins to increase when greater measurement is obtained. The final model presents an explained variance of 97.594%.

#### 3.4.4. Analysis with the Dependent Variable L2 with Compression

The parametric model shows how gender does not significantly predict changes in the compression distance measured at L2. The variables of the smoothed model including neutral axillary circumference, neutral abdominal circumference, triceps-chest skinfold thickness, and iliac crest-abdomen skinfold thickness significantly predict the measurement with compression in L2 (significant results are shown in red) (Table 7) (Appendix A). The measurement with compression in L2 decreases as the neutral axillary circumference and the skinfold thickness in the abdomen-iliac crest increase, while it increases with larger measurements obtained in the neutral abdominal circumference and in the skinfold thickness of the chest-triceps.

The final model presents an explained variance of 49.531%.

## 4. Discussion

The present study could potentially suggest that some anthropometric characteristics predict the distance from the skin to the kidney and/or peritoneum both at the L2 and L4 levels, considering the methodology applied. These findings could help physiotherapists in choosing the size of the needle to safely perform the technique perpendicularly on the iliocostalis lumborum. It is important to note that, at the L2 level, we have performed a skin-kidney measurement. Anatomically, we must emphasize that iliocostalis lumborum muscle is not in a close relationship with the kidney, that is, we will find the quadratus lumborum separating them. To produce more controlled effects on the dry needling of the iliocostalis lumborum, we recommend a smaller needle not only to avoid reaching the kidney, but also not to reach the quadratus lumborum, thus ensuring a safe puncture of the first muscle.

There are other studies that have attempted to develop a safe dry needling protocol, correlating ultrasonography measurements with anthropometric measurements, but in other muscles [13,15]. However, and in agreement with other authors [2], although correlations can be established, ultrasound imaging is a very useful tool for performing dry needling safely. It is important to either carry out an ultrasound-guided procedure in which we can always see the needle in the ultrasound image, thus controlling its location, or to carry out a sweep of the area and some quick measurements of the structure we want to puncture—in this case, the iliocostalis lumborum—and those we do not want to reach.

The chest-triceps skinfold thickness variable was found to be able to predict all the measurements conducted in this study. In this way, the skin-kidney/peritoneum distance increases when the skinfold thickness values are higher. This is consistent since a higher value in skinfold thickness would indicate a higher proportion of fat and, therefore, a greater distance between the skin and the underlying tissues. This justification has also been made by Valera-Calero et al. [13] in reference to the BMI variable, which predicted the depth of the pleura and rhomboids in their study. However, in the present study, the BMI could only predict the measurements made at the L4 level (with and without compression).

The abdomen-iliac crest skinfold thickness could also predict all the measurements; however, in the L4 measurement without compression and L2 with compression, the distance increases as the abdomen-iliac crest skinfold thickness decreases, while in the L4 with compression and L2 without compression, the distance increases with an increase in skinfold thickness. Something similar occurs with the neutral abdominal circumference variable: It could predict the measurements in such a way that when their values increase, a greater skin-kidney distance is expected at the level of L4 with and without compression and at the level of L2 with compression. However, at the level of L2 without compression, there is a decrease in the distance with higher values of the neutral abdominal circumference.

Although this study has shown interesting results, some limitations should be acknowledged. First, the sample was based on healthy subjects and the conclusions must be adapted to symptomatic population. Second, skin-kidney/peritoneum and skin-muscle measurements were taken in an exact location, but these could not be the locations of symptomatic subjects who come to clinics; however, taking them at L2 and L4 levels reduce this limitation. In addition, this study did not consider the distance between the spinous process and the exact point where the probe was located. Further research is needed that takes into account all these limitations and calculates the inter-examinator reliability.

## 5. Conclusions

The present study provides insight into the anthropometric features and the distances of the skin-kidney/peritoneum when dry needling is performed in the iliocostalis lumborum. The neutral abdominal circumference, chest-triceps skinfold thickness, and abdomen-iliac crest skinfold thickness are the most relevant features to predict this distance. In addition, no differences have been found between the right and left side measurements, nor in the measurements with and without compression. Our findings could help clinicians choose the right needle length to perform a dry needling technique in the iliocostalis lumborum muscle, considering the methodology, but further research is needed to describe a protocol.

## Figures and Tables

**Figure 1 healthcare-10-02470-f001:**
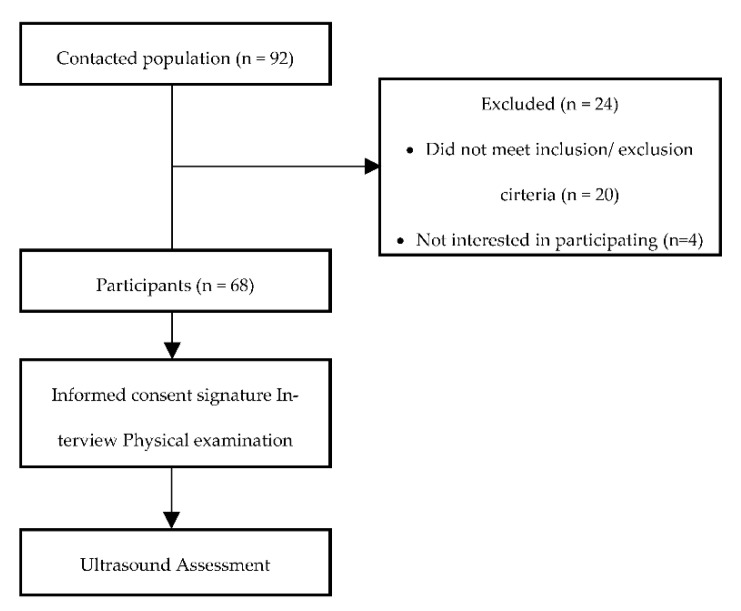
Flowchart diagram.

**Figure 2 healthcare-10-02470-f002:**
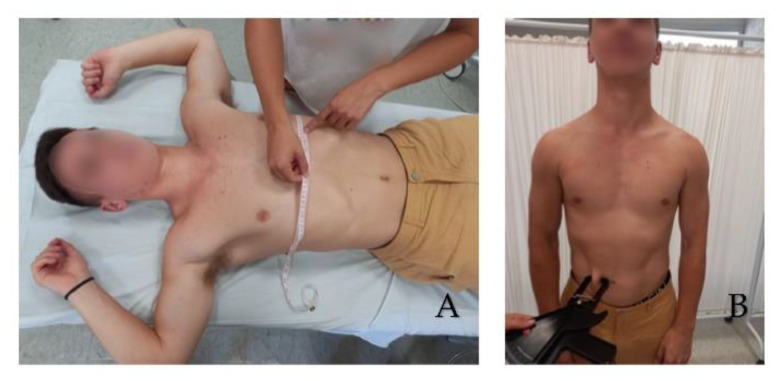
Thorax Circumference in maximum inspiration at the level of the xiphoid process (**A**). Abdominal skinfold test (**B**).

**Figure 3 healthcare-10-02470-f003:**
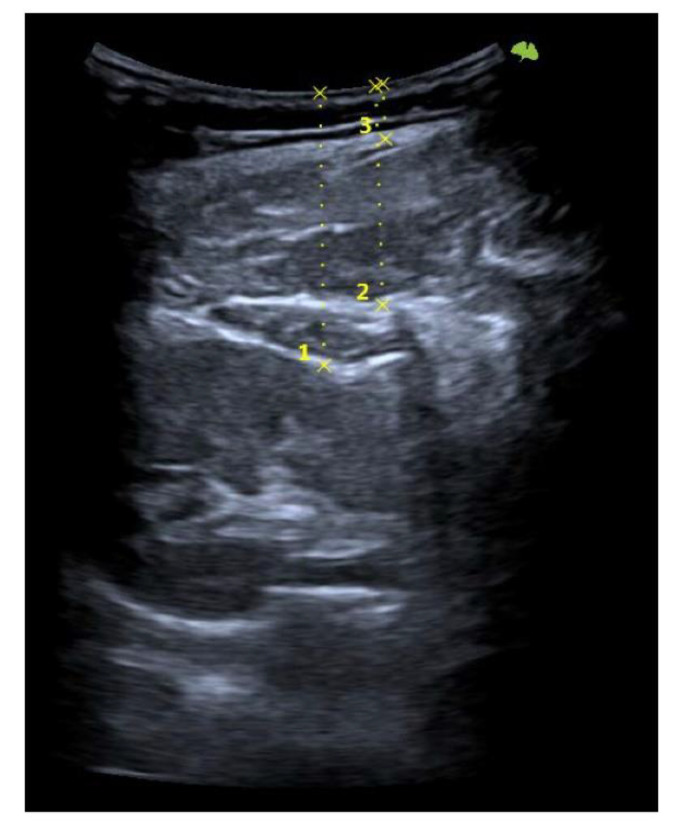
Ultrasonography image. Short-axis measurements in L2 level, left side at rest. Skin-kidney (1), skin-iliocostalis lumborum bottom edge (2), and skin-iliocostalis lumborum top edge (3).

**Figure 4 healthcare-10-02470-f004:**
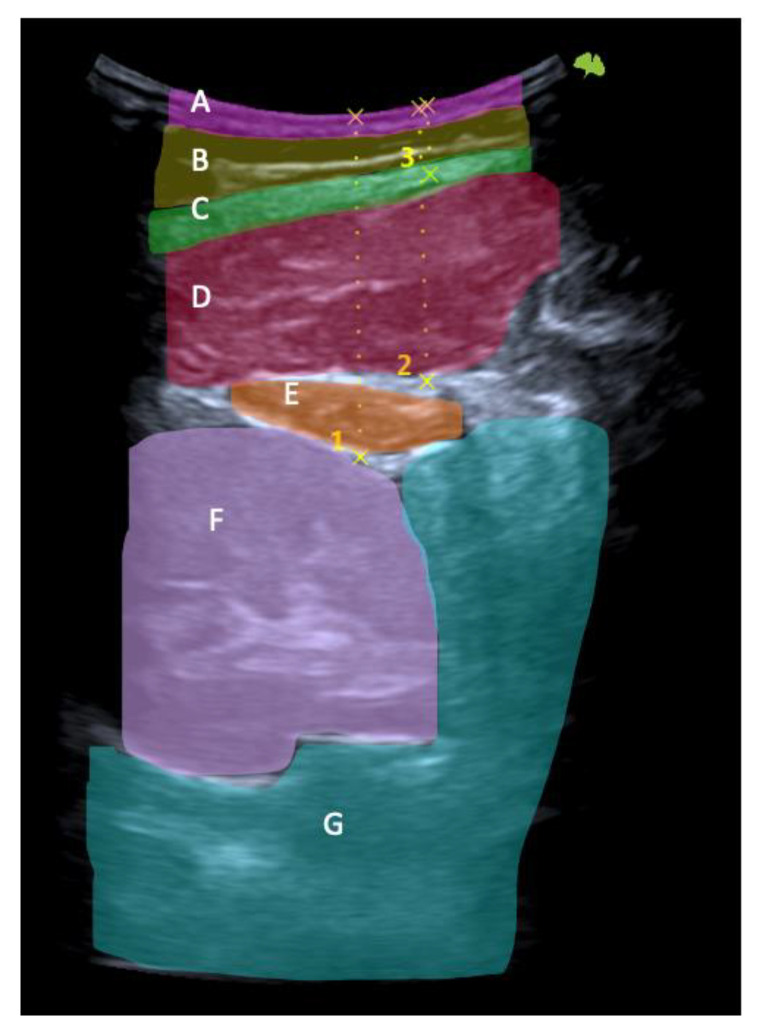
Ultrasonography image. Short-axis view: skin (**A**), fat (**B**), thoracolumbar fascia (**C**), iliocostalis muscle (**D**), quadratus lumborum muscle (**E**), kidney (**F**), and peritoneum (**G**). Skin-kidney (1), skin-iliocostalis lumborum bottom edge (2), and skin-iliocostalis lumborum top edge (3).

**Figure 5 healthcare-10-02470-f005:**
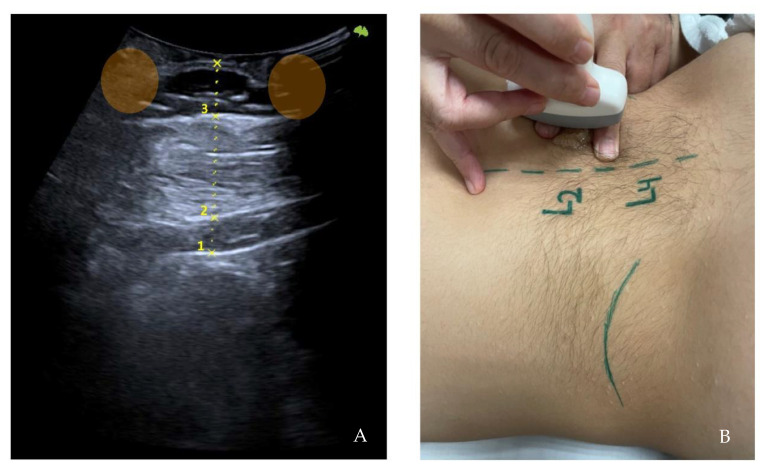
Ultrasonography image. Long-axis measurements in L2 level, left side at rest. Skin-kidney (1), skin-iliocostalis lumborum bottom edge (2), and skin-iliocostalis lumborum top edge (3) with the finger’s acoustic shadow painted (orange) (**A**). Moment of the sampling technique (**B**).

**Table 1 healthcare-10-02470-t001:** Clinical and demographic characteristics of the participants.

N		68
Gender n, (%)	Men	31 (45.6)
	Woman	37 (54.4)
Age		28.26 ± 11.03
Dominance, n (%)	Right	61 (89.7)
	Left	7 (10.3)
Weight (kg)		67.60 ± 13.38
Height (cm)		169.57 ± 9.84
Body mass index		23.33 ± 3.05
Neutral axillary circumference		93.91 ± 10.04
Neutral xiphoid circumference		83.99 ± 9.85
Neutral abdominal circumference		81.63 ± 10.57
Inspiration axillary circumference		96.22 ± 10.30
Expiration axillary circumference		93.41 ± 10.15
Inspiration xiphoid circumference		86.75 ± 9.69
Expiration xiphoid circumference		83.01 ± 10.20
Inspiration abdominal circumference		81.54 ± 10.55
Expiration abdominal circumference		81.24 ± 10.57
Chest-triceps skinfold thickness		13.47 ± 7.39
Abdomen-iliac crest skinfold thickness		14.66 ± 9.35
Thigh skinfold thickness		24.40 ± 11.32
Fat percentage		18.41 ± 7.57

Data expressed as mean±standard deviation or as absolute and relative values (%).

**Table 2 healthcare-10-02470-t002:** Intra-observer reliability.

	ICC (IC95%)	^a^*p* Valor	ICC Categorical	SEM (IC95%)
L2 right without compression: skin-kidney	0.92 (0.792, 0.977)	<0.001	Excellent	0.206 (0.114, 0.298)
L2 right with compression: skin-kidney.	0.96 (0.889, 0.989)	<0.001	Excellent	0.161 (0.101, 0.221)
L4 right without compression: skin-peritoneum.	0.951 (0.869, 0.986)	<0.001	Excellent	0.202 (0.127, 0.277)
L4 right with compression: skin-peritoneum.	0.962 (0.895, 0.99)	<0.001	Excellent	0.177 (0.125, 0.229)
L2 left without compression: skin-kidney.	0.952 (0.864, 0.987)	<0.001	Excellent	0.144 (0.086, 0.202)
L2 left with compression: skin-kidney.	0.948 (0.86, 0.986)	<0.001	Excellent	0.129 (0.089, 0.17)
L4 left without compression: skin-peritoneum	0.947 (0.858, 0.985)	<0.001	Excellent	0.201 (0.124, 0.278)
L4 left with compression: skin-peritoneum.	0.979 (0.94, 0.994)	<0.001	Excellent	0.139 (0.092, 0.187)

ICC: intraclass correlation coefficient; SEM: standard error of measurement; IC95%: confidence interval 95%. ^a^ significant *p* < 0.05.

**Table 3 healthcare-10-02470-t003:** Reliability with compression vs. without compression.

	ICC (IC95%)	^a^*p* Valor	ICC Categories	SEM (IC95%)
L2 right: skin-kidney without vs. with compression.	0.402 (−0.057, 0.82)	0.13	Poor	0.284 (0.081, 0.488)
L4 right: skin-peritoneum without vs. with compression	0.75 (0.083, 0.938)	0.015	Moderate	0.36 (0.111, 0.608)
L2 left: skin-kidney without vs. with compression	0.696 (0.054, 0.92)	0.017	Moderate	0.29 (0.132, 0.448)
L4 left: skin-peritoneum without vs. with compression	0.902 (0.244, 0.98)	0.008	Excellent	0.202 (0.066, 0.337)

ICC: intraclass correlation coefficient; SEM: standard error of measurement; IC95%: Confidence interval 95%. ^a^ significant *p* < 0.05.

**Table 4 healthcare-10-02470-t004:** Smoothed model of L4 without compression.

	EDF	df_ref_	F	^a^*p* Valor
Body mass index	1.419	2	2.147	0.048
Neutral axillary circumference	0.839	3	1.225	0.033
Neutral abdominal circumference	1.975	3	8.726	<0.001
Chest-triceps skinfold thickness	1.458	3	2.412	0.012
Abdomen-iliac crest skinfold thickness	1.000	3	4.957	<0.001

EDF: effective degrees of freedom. df_ref_: reference degrees of freedom. ^a^ significant *p* < 0.05.

**Table 5 healthcare-10-02470-t005:** Smoothed model of L4 with compression.

	EDF	df_ref_	F	^a^*p* Valor
Body mass index	1.907	3	2.873	0.009
Neutral abdominal circumference	1.997	3	10.260	<0.001
Chest-triceps skinfold thickness	0.892	3	2.684	0.003
Abdomen-iliac crest skinfold thickness	0.963	3	4.604	<0.001

EDF: effective degrees of freedom. df_ref_: reference degrees of freedom. ^a^ significant *p* < 0.05.

**Table 6 healthcare-10-02470-t006:** Parametric model of L2 without compression.

	Coefficient	Standard Error	t	^a^*p* Valor
(Intercept)	16.019	0.408	39.227	<0.001
Gender (female)	4.387	0.695	6.310	<0.001

^a^ significant *p* < 0.05.

**Table 7 healthcare-10-02470-t007:** Smoothed model with the dependent variable L2 with compression.

	EDF	df_ref_	F	^a^*p* Value
Neutral axillary circumference	0.863	3	1.596	0.017
Neutral abdominal circumference	1.962	3	10.198	<0.001
Chest-triceps skinfold thickness	0.874	3	2.073	0.009
Abdomen-iliac crest skinfold thickness	1.000	3	4.677	<0.001

EDF: effective degrees of freedom. df_ref_: Reference degrees of freedom. ^a^ significant *p* < 0.05.

## Data Availability

Not applicable.

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
