# Peer review of "Correlation between Anthropometric and Ultrasound Measurement for Dry Needling of the Iliocostalis Lumborum Muscle with a Safety Protocol: A Cross-Sectional Observational Study"

_healthcare, 2022, doi:10.3390/healthcare10122470_

Round 1
Reviewer 1 Report
I got the manuscript for review: Anthropometric and ultrasound measurements correlation for 2 dry needling of the iliocostalis lumborum muscle with a safety 3 protocol: cross-sectional observational study. These are interesting studies and, in my opinion, very aptly beneficial for the development of modern physiotherapeutic methods.
60 The information on internal organs (kidneys) should be elaborated on by describing the dangers of dry needling
89-97 In the manuscript, please specify one specific purpose. Objectives formulated in this way are illegible.
106 How many points did the study score in the STROBE questionnaire?
108 what is the minimum sample size for this study? Based on what parameters was the sample size calculated?
120 Was the study protocol compliant with the Helsinki Declaration?
The very precisely described research methodology deserves recognition.
Was the control group considered? (group of measurement bases in ultrasound)
204-209 This data should be in the "statistical analysis" section
264 Error estándar? No translation
The objectives of the study are too precise with such a small group size and high variability in the data
Author Response
REVIEWER 1
I got the manuscript for review: Anthropometric and ultrasound measurements correlation for 2 dry needling of the iliocostalis lumborum muscle with a safety 3 protocol: cross-sectional observational study. These are interesting studies and, in my opinion, very aptly beneficial for the development of modern physiotherapeutic methods.
60 The information on internal organs (kidneys) should be elaborated on by describing the dangers of dry needling.
Dear reviewer, thanks for the comment. It has been detailed.
89-97 In the manuscript, please specify one specific purpose. Objectives formulated in this way are illegible.
Dear reviewer, thanks for the comment. It is now specified in lines 94-97.
106 How many points did the study score in the STROBE questionnaire?
Dear reviewer, thanks for the comment. The STROBE has been filled the and attached with this review.
108 what is the minimum sample size for this study? Based on what parameters was the sample size calculated?
Dear reviewer, thanks for the comment. We have calculated it based on the parameters used by others authors that made a similar study. Detailed in lines 173-177. The minimum was 40 subjects.
120 Was the study protocol compliant with the Helsinki Declaration?
Dear reviewer, thanks for the comment. Yes, the study was carried out following all the guidelines approved by the Ethics Committee for Animal Research and Experimentation (CEI-EA) of the UAH which is based on Helsinki Declaration, detailed in lines 220-225.
The very precisely described research methodology deserves recognition.
Thank you very much for your comment.
Was the control group considered? (group of measurement bases in ultrasound).
Dear reviewer, thanks for the comment. The aim of the research was to evaluate the depth of the abdominal structures to know the potential risks due to the unknown by measuring the thickness of the different layers. The selection criteria were healthy population in an observational study design this is the reason why is not necessary a control group.
204-209 This data should be in the "statistical analysis" section.
Dear reviewer, thanks for the comment. It has been changed. See now at lines 352-357.
264 Error estándar? No translation.
Dear reviewer, thanks for the comment It has been translated, line 407
The objectives of the study are too precise with such a small group size and high variability in the data
Dear reviewer, thanks for the comment, this issue has been considered for the discussion.

Reviewer 2 Report
Remove the dot at the end of the title
The usual acronym for myofascial trigger point is (MTrP) – verify throughout manuscript.
The usual acronym for Dry needling is (DN) - check throughout the manuscript.
Insert hypothesis after objective.
"This study follows the 105 STROBE Statement" review the suitability according to STROBE equator guidelines and attach the guideline sheet to demonstrate compliance.
The information: “For this, an attempt will be made to establish the relationship between the anthropometric measurements of the subjects including age, sex, height, weight, body mass index (BMI), chest (axillary and xiphoid process) and abdomen circumference, skinfold thickness measurement, and the distance skin-kidney/peritoneum and skin-muscle measured with an ultrasound scanner. This study follows the 105 STROBE Statement (16,17)” are not Study Design. Please insert in appropriate place.
The description of the participants is superficial, the sample calculation needs to be detailed to know the minimum sample obtained and based on what information from these articles this answer was obtained.
Include selection process flowchart.
I also suggest dividing into subtopics: Population, sampling, sample, eligibility criteria.
“Ethics Committee for Animal Research and Experimentation (CEI-EA)” is this type of committee able to evaluate human studies in Spain?
What does "UAH" mean?
Quality of figure 1 and 2 need to be improved, they can be integrated. In addition, if you are going to have these 2 measures, I suggest including the others as a guideline to facilitate the replication of measures as a security protocol.
Figures 3, 4 and 5 must be integrated. In addition, I suggest putting a photo of the moment of application of the technique.
What type of transducer was used?
What is the depth established for evaluation?
Check the established significance level.
I did not check qualitative variables, I believe that the correct is categorical
Ultrasound is starting the phrase in lower case in the statistical analysis.
Multicollinearity and tolerance must be tested in the regression model.
Results are confusing, with tables that need to be organized, results and models must be better presented to facilitate understanding.
There are many words in another language, so I believe it should be submitted to an English language certified company.
Acronyms are inserted into the text without explaining the meaning.
Different ethical principles in the method and after completion need to be adjusted.
I need to better understand the results to review the discussion and conclusions.
The theme of the article is very interesting, but the manuscript needs to be revised.
Author Response
REVIEWER 2
Remove the dot at the end of the title.
Dear reviewer, thanks for the comment, it has been removed.
The usual acronym for myofascial trigger point is (MTrP) – verify throughout manuscript..
Dear reviewer, thanks for the comment, it has been adapted.
The usual acronym for Dry needling is (DN) - check throughout the manuscript..
Dear reviewer, thanks for the comment, it has been translated.
Insert hypothesis after objective..
Dear reviewer, thanks for the comment, it has been inserted at lines 97-102.
"This study follows the 105 STROBE Statement" review the suitability according to STROBE equator guidelines and attach the guideline sheet to demonstrate compliance.
Dear reviewer, thanks for the comment the STROBE cheklist has been reviewed and attached with this revision.
The information: “For this, an attempt will be made to establish the relationship between the anthropometric measurements of the subjects including age, sex, height, weight, body mass index (BMI), chest (axillary and xiphoid process) and abdomen circumference, skinfold thickness measurement, and the distance skin-kidney/peritoneum and skin-muscle measured with an ultrasound scanner. This study follows the 105 STROBE Statement (16,17)” are not Study Design. Please insert in appropriate place.
Dear reviewer, thanks for the comment. This part belongs to the study design and we have been reviewing and consider this would be the best section where it must be at lines 159-162, please take a sight.
The description of the participants is superficial, the sample calculation needs to be detailed to know the minimum sample obtained and based on what information from these articles this answer was obtained.
Dear reviewer, thanks for the comment, the description has been enlarged lines 175-177.
Include selection process flowchart.
Dear reviewer, thanks for the comment, a flowchart has been added in line 172
I also suggest dividing into subtopics: Population, sampling, sample, eligibility criteria.
Dear reviewer, thanks for the comment. It has been done.
“Ethics Committee for Animal Research and Experimentation (CEI-EA)” is this type of committee able to evaluate human studies in Spain?
Yes, it is.
The Ethics Committee for Animal Research and Experimentation (CEI-EA) of the University of Alcalá (UAH) has the mission of informing about the ethical aspects of the research projects of the UAH and advising the University Community on ethical issues in the production of knowledge and in the publication of research results. Any research project carried out with human beings, animals, biological agents and genetically modified organisms.
This is the link of the CEI-EA:
https://www.uah.es/es/investigacion/servicios-para-el-investigador/comite-de-etica-de-investigacion-y-experimentacion-animal/?buscadoreticafield-1=faqs%2FComite-de-Etica-de-la-Investigacion-animal%2FActividades-evaluables-y-autorizadas-por-el-CEIEA%2F&numfield=1&searchaction=search&searchPage=1&submit=Buscar#presentacion
What does "UAH" mean?
Universidad de Alcalá de Henares (University of Alcalá)
Quality of figure 1 and 2 need to be improved, they can be integrated. In addition, if you are going to have these 2 measures, I suggest including the others as a guideline to facilitate the replication of measures as a security protocol.
Dear reviewer, thanks for the comment. It has been enhanced, now figure 2.
Figures 3, 4 and 5 must be integrated. In addition, I suggest putting a photo of the moment of application of the technique.
Dear reviewer, thanks for the comment. The figures have been enhanced and integrated.
What type of transducer was used?
A Vinno model E35 ultrasound machine with a convex probe with a footprint of 40 mm was used as an exploration and measurement tool. See at line 249
What is the depth established for evaluation?
Dear reviewer, thanks for the comment. The depth is the distance since the surface of the skin till the structure of interest (like peritoneum, kidney…) and it varies with each subject as we would want to optimize the image.
Check the established significance level.
Dear reviewer, thanks for the comment. It has been checked.
I did not check qualitative variables, I believe that the correct is categorical
Dear reviewer, thanks for the comment. It has been corrected. Line 347
Ultrasound is starting the phrase in lower case in the statistical analysis.
Dear reviewer, thanks for the comment. It has been modified
Multicollinearity and tolerance must be tested in the regression model. Results are confusing, with tables that need to be organized, results and models must be better presented to facilitate understanding.
Dear reviewer, thanks for the comment. We have reorganized them.
There are many words in another language, so I believe it should be submitted to an English language certified company.
Dear reviewer, thanks for the comment, the text has been reviewed and translated those in Spanish.
Acronyms are inserted into the text without explaining the meaning.
Dear reviewer, thanks for the comment. The acronyms has been explained.
Different ethical principles in the method and after completion need to be adjusted.
Dear reviewer, thanks for the comment. It has been adjusted.
I need to better understand the results to review the discussion and conclusions.
Dear reviewer, thanks for the comment. Hope to know about.
The theme of the article is very interesting, but the manuscript needs to be revised.
Dear reviewer, thanks for the comment. Hope this new version have got enough revision and enhancement.
